# Xeroderma Pigmentosum Type C Primary Skin Fibroblasts Overexpress HGF and Promote Squamous Cell Carcinoma Invasion in the Absence of Genotoxic Stress

**DOI:** 10.3390/cancers16193277

**Published:** 2024-09-26

**Authors:** Sahar Al-qaraghuli, Yannick Gache, Maria Goncalves-Maia, Damien Alcor, Elodie Muzotte, Walid Mahfouf, Hamid-Reza Rezvani, Thierry Magnaldo

**Affiliations:** 1INSERM U1081–CNRS UMR7284-UNS, CEDEX 02, F-06107 Nice, France; sahar.alqaraghuli@gmail.com (S.A.-q.);; 2Faculté de Médicine, 2ème étage, CNRS UMR 6267—INSERM U998—UNSA, F-06107 Nice Cedex 2, France; 3INSERM U1065, C3M, Microscopy Facility, F-06200 Nice, France; 4BRIC, UMR 1312, Inserm, Université de Bordeaux, F-33076 Bordeaux, France; 5Centre de Référence pour les Maladies Rares de la Peau, CHU de Bordeaux, F-33000 Bordeaux, France

**Keywords:** xeroderma pigmentosum, fibroblasts, HGF/SF, squamous cell carcinoma

## Abstract

**Simple Summary:**

Xeroderma pigmentosum (XP) is a very rare recessive disease caused by the incapacity to resolve ultraviolet-induced DNA lesions through Nucleotide Excision Repair (NER). Some XP patients suffer from aggressive skin carcinoma and melanoma at a very early age (<8). Our previous results show that primary XP fibroblasts isolated from healthy, non-photo-exposed skin negatively impact the extracellular matrix and the activation of the innate immune system. Here, we show for the first time that XP group C primary fibroblasts also play a major role in cancer cell invasion ex vivo and in vivo through the overexpression of Hepatocyte Growth Factor/Scatter Factor (HGF/SF) in the absence of genotoxic attacks. Using inhibitors of the activation of the HGF/SF pathway counteracted the effects of XP fibroblasts on the growth of cancer cells, suggesting new perspectives in the care of XP patients.

**Abstract:**

Xeroderma pigmentosum (XP) is a very rare recessive disease caused by the incapacity to resolve ultraviolet-induced DNA lesions through Nucleotide Excision Repair (NER). Most XP patients suffer from aggressive skin carcinoma and melanoma at a very early age (<8). Our previous results showed that primary XP fibroblasts isolated from healthy (non-photo-exposed) skin negatively impact the extracellular matrix and fail to activate the innate immune system. Here, we show for the first time that XP-C fibroblasts also play a major role in cancer cell invasion ex vivo and in vivo through the overexpression of Hepatocyte Growth Factor/Scatter Factor (HGF/SF) in the absence of genotoxic attacks. The use of inhibitors of the activation of the HGF/SF pathway counteracted the effects of XP fibroblasts on the growth of cancer cells, suggesting new perspectives in the care of XP patients.

## 1. Introduction

Skin carcinomas occupy the first place in human cancers (about 30%) in terms of incidence (https://www.cancer.gov/types/skin; 15 May 2024) [1,2]. DNA repair mechanisms are essential to ensuring genetic maintenance and avoiding deleterious mutations that can be related to aging and cancer development. Among the prototype diseases associated with these harmful processes, xeroderma pigmentosum (XP) is very rare (about 1/500,000 births in European countries), and is inherited as a recessive and autosomal disorder. The XP syndrome is due to a failure in the Nucleotide Repair Mechanism (NER) that removes bulky DNA lesions, notably, pyrimidine dimers (i.e., UV-induced DNA lesions introduced in adjacent pyrimidines in the same DNA strand: cyclobutane pyrimidine dimers (CPDs), 6-4 pyrimidine-pyrimidone photo products (6-4 PP), and benzopyrene adducts [3,4]), leading to a high susceptibility towards cutaneous cancers (CCs), notably squamous cell carcinomas (SCCs) [5]. Up to now, seven groups of XP genetic complementation (XP-A to XP-G) associated with the disease have been identified. Among them, XP-C is mostly prevalent globally, accounting for about 50% of all XP patients. 

Surprisingly, XP patients from the general population develop about 5 times more BCCs than SCCs, while this ratio is reversed in XP-C patients [6]. In this respect, studies from Dr. De Feraudy revealed for the first time that the *XPC* gene is inactivated in a significant part of skin SCCs [5] and this gene controls SCC development, as was suggested by Dong and colleagues [7].

Altogether these observations suggest that XPC behaves both as a care- and gate-keeper factor. In addition, as was already shown in the case of Carcinoma-Associated Fibroblasts (CAFs) [8] using organotypic skin cultures, our results revealed increased contractility of the extracellular matrix (ECM) while XP-C fibroblasts populated the dermal equivalent [9]. 

XP-C fibroblasts also fail to express the activator of natural killer lymphocytes, CLEC2A (C-Type Lectin Domain Family 2 Member A), leading to CC invasion in immunocompetent organotypic skin cultures [10]. Altogether, these observations suggest that XPC behaves as both a care- and gate-keeper factor. The presumptive roles of XPC in controlling genetic expression via its participation in either the genetic or epigenetic control of gene expression deserves further studies. In addition to these observations linking XPC with the control of gene expression, we show here that primary XP-C fibroblasts spontaneously overexpress the mitogenic/motogenic factor HGF/SF, leading to the activation of mesenchymal–epithelial transition factor (cMET), signal transducer and activator of transcription 3 (STAT3), and c-Jun NH2-terminal kinase (JNK) downstream pathways in SCC cells ex vivo. The blockade of HGF–cMET interactions resulted in the inhibition of both STAT3 and JNK phosphorylation, together with SCC invasion ex vivo. These results pave the way for better clinical care of XP-C patients through anti-HGF approaches.

## 2. Materials and Methods

### 2.1. Cell Culture

Human primary fibroblasts and keratinocytes were isolated from healthy, non-photo-exposed skin biopsies from either control individuals (WT) or XP-C patients (Appendix A). Written informed consent was provided by patients or their relatives in accordance with the Declaration of Helsinki and French law. This study was approved by the Institutional Review Board of the University Institute of Hematology (IUH; Saint-Louis Hospital, Paris, France), the French Agency of Biomedicine (Paris, France) (Arrêté N° 2001/904 and Ref: AG08-0321 GEN of 27 September 2008 www.agence-biomedecine.fr/Genetique), and the European Commission “Geneskin: Genetics of human genoder-matosis” (Brussels, Belgium).

SCC-13 and 3T3-J2 cells were a generous gift from Dr. J. Rheinwald [11] and were cultured as described in [12]. γ-irradiated 3T3-J2 feeder cells were removed with phosphate-buffered saline–ethylene diamine tetra-acetic acid (PBS-EDTA) 0.02% for 5 min. SCC13 cells were then re-fed for 48 h in culture medium (Fibroblast Adenine DMEM, FAD; 0.5% fetal calf serum (FCS)) and cultured for 1h30 in fibroblast-conditioned culture medium supernatants (CMs). Human fibroblasts were seeded at a density of 6.6 10^3^ cells/cm^2^, cultured for 72 h in cFAD with 10% FCS, washed in PBS, and cultured for 48 h in FAD medium with 0.5% FCS. The collected CM was then spun for 5 min at 5000 g. SCC13 cells were then seeded at a density of 4.10^3^ cells/cm^2^ in cFAD for 48 h to reach about 50% confluency. The fibroblast CM was supplemented with 0.66 ng/mL of the anti-HGF blocking monoclonal antibody (MAB294) for 1 h at 37 °C prior to the transfer to SCC13 cells.

### 2.2. Scratch Assay

γ-3T3-J2 cells were seeded at a density of 15.10^3^ cells/cm^2^ and cultured for 24 h in cFAD. SCC13 cells were seeded at a density of 14.10^4^ cells/cm^2^ in 6-well plates in g3T3-J2-CM for 24 h before culturing in FAD with 0.5% FCS for 24 h. The cells were then scratched using 200 µL tips, washed in PBS, and incubated ± WT or XP-C fibroblast CM. SCC13 cells were then grown ± 10 µg/mL of MMC for 2 h prior to scratching and the analysis was performed using a live epi-fluorescence microscope (Carl Zeiss Microscopy, Jena, Germany) for 10 h and quantified using the EpiDepth software [10].

### 2.3. Contractile Property Measurements and Invasion Assays in Organotypic Skin Cultures

A total of 25.10^3^ fibroblasts (either WT or XP-C) were embedded in 100 μL of a mix of rat tail collagen I (#354249; BD Biosciences, Oxford, UK) and Matrigel^®^ (#354234; BD Biosystems), at concentrations of 4.6 mg/mL and 2.2 mg/mL, respectively [13,14]. The cells/ECM mix was then seeded in triplicate into 96-well plates. After 1 hour at 37 °C, the gels were overlaid with 100 μL of FAD, 0.5% FCS ± 0.66 ng/mL of HGF/SF neutralizing monoclonal antibody (#MAB294). After 24 h, the mean diameters of the gels were measured using the ImageJ software.

### 2.4. Histology and Immunostaining

Samples were fixed in 10% neutral buffered formalin, embedded into paraffin, and sectioned for routine histology staining (hematoxylin–eosin, H&E, Hong Kong). For indirect immune-labeling of keratin 14 (K14, AbCam NCL-LL002, 1/20), paraffin sections were de-waxed and subjected to heat-induced epitope retrieval after boiling in a citrate buffer (5 min), blocked, and permeabilized in 0.1% Triton X-100 with 3% bovine serum albumin (5 min). Immunostaining was performed using anti-K14 mAb (Abcam, Cambridge, UK) and rabbit anti-mouse Alexa Fluor 594 (Invitrogen, Carlsbad, CA, USA, 1/2000).

For mouse skin immunofluorescence staining, 4 μm paraffin-embedded tissue sections were deparaffinized in xylene and rehydrated in progressively decreasing ethanol concentrations. Antigen retrieval was performed by heating in sodium citrate (pH 6.0) and then saturated using PBS-5% SVF. The sections were then incubated overnight at 4 °C using the first primary antibody directed against XPC (sc-74410, Santa Cruz, 1/50) and then incubation with Alexa Fluor 568 (#16-237, Normal Rabbit IgG, Alexa Fluor™ 488 conjugate; 1/3000)-conjugated secondary antibody for 1 h at room temperature. After de-hybridization of the primary antibody, the tissue sections were then incubated overnight at 4 °C with a second anti-vimentin antibody (M0725, Dako, 1/300), followed by incubation with a conjugated secondary antibody. Nuclei were then counterstained with DAPI for 10 min at room temperature before mounting in Fluoromount^TM^ (aqueous mounting medium, Sigma, Kanagawa, Japan). The sections were observed and imaged using an epi-fluorescence microscope (Nikon Eclipse, Tokyo, Japan).

### 2.5. Protein Extraction and Analyses

Cells were lysed on ice in TLB buffer (Tris 20 mM pH 7.5, NaCl 137 mM, EDTA 2 mM pH 7.5, Triton X-100 1%, Na PPi 2 mM, glycerol 10%) with protease inhibitors (#11836170001; Roche, Mannheim, Germany) and phosphatase inhibitors (#04906837001; Roche). Proteins (30 µg) were separated by 8% SDS-PAGE and transferred onto polyvinyl difluoride (PVDF) membranes (Millipore, Billerica, MA, USA). The membranes were saturated with 5% bovine serum albumin, 10 mM Tris-HCl (pH 7.5), 500 mM NaCl, and 0.1% Tween 20, probed with specific primary antibodies and then with HRP-labeled secondary antibodies. Immunodetection was performed using the Luminata Crescendo Western HRP Substrate (Millipore Corporation, Billerica, MA, USA) and Fusion Solo software (Vilber Lourmat, Eberhardzell, Germany) or using exposure on Amersham films (GE Healthcare Limited, Bucinghamshire, UK). The membranes were then stripped and re-probed using an anti-GAPDH mouse mAb.

### 2.6. RNA Extraction and Quantitative RT-PCR

Total RNA was extracted using the RNeasy Mini kit (Qiagen, Hilden, Germany). Reverse transcription was performed from 1 µg RNA using the Superscript II Reverse Transcriptase (Roche Applied Science, Basel, Switzerland) and hexameric random primers (Invitrogen, Carlsbad, CA, USA). Real-time PCR was carried out on cDNAs using the primers listed in Appendix A. For Taqman primers, RT-q-PCR was carried out using the 7900 Fast Real-Time PCR system (Applied Biosystems, Forster City, CA, USA). *GAPDH, PPIA, RPLO1* were used as housekeeping genes. The results were normalized using the GeNorm software. For the SYBR green analyses, RT-q-PCR was carried out using Fast SYBR Green Master Mix as advised by the manufacturer (#18064-014; Applied Biosystems, Foster City, CA, USA) and the StepOnePlus^TM^ Real-Time PCR System (Applied Biosystems). *GAPDH* and *SB34* primers were used as the most valuable control housekeeping gene sequences.

### 2.7. Spheroid Assay

Cultured SSC13-GFP (Green Fluorescent Protein) cells and fibroblasts were dissociated by trypsin/EDTA treatment, spun, and then resuspended at a density of 10^6^ cells/mL in cFAD. Fibroblasts were then pre-stained for 30 min using either the nuclear cell tracker NucBlue-Live ReadyProbes^R^ reagent (# R37605; Life Technologies, Eugene, OR, USA) or the NucRed-Live ReadyProbes^R^ reagent (# R37106; Life Technologies, Carlsbad, CA, USA). A total of 25.10^3^ of both SCC13-GFP and WT or XP-C fibroblasts were then mixed in a 50 μL drop of spheroid mix on a 60 mm culture dish, and incubated for 72 h at 37 °C [15]. The spheroids were then transferred onto 14 mm glass-bottomed cell culture dishes (P35GC-1.5-14-C, MatTek corporation, Ashland, Wilmington, DE, USA), incubated for 15 to 30 min at 37 °C and then for 24 h in cFAD at 37 °C before confocal microscopy (Nikon A1R, Leica AM-TIRF).

### 2.8. HGF Enzyme-Linked Immunoassay

Culture supernatants were collected after 48 h of culture in 0.5% FAD serum. The amount of secreted HGF was analyzed using the Human HGF ELISA Kit (# ELH-HGF-001, RayBiotech, Norcross, GA, USA) according to the manufacturer’s instructions.

### 2.9. Transduction of XP-C Fibroblasts and Fluorescence-Activated Cell Sorting

SCC13-GFP cells were obtained after transduction of SCC-13 cells with the pCMMP-CD24-IRES-GFP vector as described in [16]. XP-C fibroblasts were infected with the retroviral vector pCMMP CD24-IRES-XPC [12]. GFP+ cells were sorted out using a FACS ARIA instrument (BD Biosciences; San Jose, CA, USA). 

### 2.10. Quantitation of ROS Accumulation

Primary fibroblasts were seeded at a density of 6700 cells/cm^2^, grown for 48 h, and incubated in 1 µM chloromethyl-2′,7′-dichlorodihydrofluorescein diacetate (CM-H_2_DCFDA, # C6827, Molecular Probes, Invitrogen, Bend, OR, USA) in Hank’s buffered salt solution for 30 min at 37 °C. After 2 washes in HBSS, the cells were UVA irradiated at 4 000 mJ/cm^2^ and then incubated for 30 min at 37 °C in HBSS. As a positive control of the pro-oxidant effects of UVA, fibroblasts were incubated in HBSS medium containing 100 µM H_2_O_2_ at 37 °C for 30 min. After dissociation and resuspension in HBSS with 10% SVF, the fluorescence intensity was measured by cytometry.

### 2.11. Animals and Experimental Protocol

NOD/Shi-SCID IL2Rγ-null mice (NSG) were bred in standard conditions compliant with the regulations and housed in a pathogen-free animal facility. All experiments were carried out with the approval of the Bordeaux University Animal Care and Ethical Committee. Thirty 8–10-week-old females were randomly divided in 6 groups of five mice before they were given a subcutaneous injection in the right and left flanks. For each condition, 100 µL of a cell suspension containing 7.10^5^ SCC13 tumor cells were injected with 7.10^5^ of either WT or XP-C fibroblasts in high concentration Matrigel^®^ (Corning, Tewksbury, MA, USA). Twice a week, the tumor volume (V) was calculated as follows: V = (width)^2^ × length/2. After 2 months, the mice were sacrificed for tumor extraction and further analyses.

### 2.12. Statistical Analysis

Student’s *t*-tests were performed for the statistical analysis of the invasion assay, gel contraction, quantitative PCR, ELISA, and Western blot results. *** *p* < 0.001, ** *p* < 0.01, * *p* < 0.05. Error bars show the +S.D.

## 3. Results

### 3.1. XP-C Primary Fibroblasts Promote Invasiveness of SCC Ex Vivo

In organotypic skin cultures, neither SCC nor WT keratinocytes exhibited a significant invasive capacity in the dermal compartment containing WT fibroblasts (Figure 1A). In contrast, the XP-C fibroblasts in dermal equivalents resulted in the massive invasion of SCC cells (Figure 1A,B). K14 cytokeratin labeling confirmed the epithelial origin of the invading cells (Figure 1C). Spheroid assays mixing SCC13-GFP cells with either red WT or red XP-C fibroblasts [8,15] showed more cancer cell protrusions in the presence of XP-C vs. WT fibroblasts (Figure 1D). Time-lapse analyses showed that, like CAFs, XP-C fibroblasts stepped ahead of SCC cell protrusions (Figure 1E).

### 3.2. XP-C Fibroblasts Accelerate Scratch Closure of SCC Cells

Whether the pro-invasive activity of primary XP-C fibroblasts was due to mitogenic and/or motogenic effects on SCC cells was analyzed by ex vivo scratch assays in the presence or absence of mitomycin C (MMC). In XP-C fibroblast CM, the scratch closure of SCC cells was almost completed 7h30 post-injury (PI, 70 to 90%) compared to WT fibroblast CM with only 50% closure at the same time point (Figure 2A). At 10h30 PI, the slope of scratch closure of the cancer cells in the presence of XP-C fibroblast CM decreased slowly to reach 95% while in WT fibroblast CM, the scratch closure of the SCC cells was limited to 65%. MMC treatment of SCC cells led to similar scratch closure rates in both WT and XP-C fibroblast CMs (65 to 75% at 10h30 PI (Figure 2B)). Thus, XP-C fibroblast CM seemed to increase SCC scratch cell closure ex vivo through the secretion of mitogenic rather than motogenic signals. However, XP-C fibroblast CM did not significantly affect the numbers of SCC cells in the G1, G2, or S phases compared to WT fibroblast CM (Appendix A). Also, the amounts of cyclin B1, a master protein in G2-M cell cycle progression, were similar in SCC cells cultured in WT vs. XP-C CMs (Appendix A). 

### 3.3. XP-C Fibroblasts Overexpress HGF in a Cell-Autonomous Manner

The analysis of mRNA amounts of factors known to be expressed by CAFs, i.e., SDF1α, IL6, FGF7, LIF1, HGF/SF, TGFβ, OSM, CXCR7, GDF15, CXCL1, and EGF [17], indicated significantly higher HGF/SF mRNA and protein levels secreted by XP-C vs. WT fibroblasts (Figure 3A,B). The mRNA levels of LIF1, SDF1α/CXCL12, and EGF were only slightly increased in primary XP-C vs. WT fibroblasts (1.5- to 2-fold, data not shown). The levels of α-SMA, a marker of CAFs [18,19], did not show any difference between WT vs. XP-C fibroblasts (data not shown). Surprisingly enough, in genetically complemented XP-C fibroblasts [12], HGF mRNA and protein secretion (Appendix A–C) were not normalized despite the recovery of normal NER levels (data not shown). 

### 3.4. HGF/SF Overexpression in XP-C Fibroblasts Activates cMET and Downstream STAT3 and JNK Signaling Pathways

p-cMET was significantly increased in SCC cells grown in CMs of XP-C vs. WT fibroblasts (Figure 3C). The activity of signaling pathways activated by HGF/SF were then measured by assessing the phosphorylation of STAT3, JNKs, ERKs, and P38 in SCC cells exposed to CMs of XP-C vs. WT fibroblasts (Figure 3D–G). p-STAT3 and p-JNK1/2 levels were significantly increased in SCC cells grown in XP-C fibroblast CM (Figure 3D,E), while p-ERK1/2 and p-P38 remained unchanged (Figure 3F,G).

### 3.5. Inhibition of cMet Activation Counteracts Increased Invasiveness of SCC Cells Due to XP-C Fibroblasts 

Adding a human HGF-neutralizing antibody in the CM of XP-C fibroblasts resulted in a significant decrease in p-cMET (Figure 4A), p-STAT3, and p-JNK levels in treated SCC cells (Figure 4B). In addition, the contraction of dermal organoids containing XP-C vs. WT primary fibroblasts was significantly reduced in HGF/SF blocking antibody CM (Figure 5A). Additionally, in the presence of HGF/SF antibodies, SCC cell invasion (Figure 5B) was significantly attenuated.

### 3.6. XP-C Fibroblasts Promote Cancer Cell Invasion In Vivo

To address the influence of XP-C fibroblasts on the invasiveness of cancer cells in vivo, we performed orthotopic xenografts in mice using a mix of WT and SSC keratinocytes together with either WT or XP-C fibroblasts. Cancer cell invasion was substantially increased in the presence of XP-C vs. WT fibroblasts (Figure 6A). XP-C fibroblasts also displayed a strong propensity to accumulate in vivo and aligned with the increased SCC cell invasion. These data further confirmed the ex vivo analyses (Figure 6B), suggesting an attractant role of XP-C primary fibroblasts towards cancer cell invasion.

## 4. Discussion

Stroma–epithelium interactions are known to impact pertinent genome expression in non-cancerous cells neighboring the cancer mass, and hence promoting survival, in situ growth, and distal invasion. For the first time, we report that primary fibroblasts isolated from non-photo-exposed healthy skin from XP-C patients overexpress HGF/SF at significantly higher levels than WT fibroblasts ex vivo. In this study, we used numerous cells from different XP-C patients with various ages and clinical data (Appendix A). As showed in numerous studies with sporadic human cells [20,21], XP-C fibroblast CM activated cMET phosphorylation in SCC cells. Other ex vivo experiments showed that cMET activation led to signaling kinase cascades including ERK, P38, JNK, and the direct activation of the STAT3 transcription factor, hence contributing to cell growth and invasion in the context of malignancy [22]. Here, the invasion of SCC cells in XP-C CM was significantly increased with the activation of p-cMET, p-STAT3, and p-JNK and significantly decreased in the presence of HGF/SF inhibitors.

CAFs exhibit diverse functions linked to the secretion of cytokines and ECM proteinases, affecting cancer cell proliferation, propagation, and tissue angiogenesis, constituting a discouraging prognostic in vivo criterion [23]. XP-C fibroblasts also increased the contraction of human dermal organoids and induced SCC cancer cell invasion in organotypic skin cultures as well as in xenograft models (Figure 6). However, in contrast to previous studies showing that MMC blockade of DNA replication of SCC cells did not correlate with seminal studies showing that HGF secretion by mesenchymal cells resulted in mitogenic effects in hepatocytes [22,24], our observations further supported that XP-C fibroblasts spontaneously exhibit a CAF-like phenotype by promoting ECM remodeling and cancer cell invasion ex vivo and in vivo without promoting obvious changes in SCC cells’ cell cycle. 

In organotypic skin mounts, a blockade of HGF activity counteracted XP-C-fibroblast-led invasive and contractile properties, similar to what has been described in the presence of CAFs [8]. The spheroid assays also indicated that, like CAFs, XP-C fibroblasts behaved as leader cells by “pulling” SCC cells ex vivo. However, despite their invasive behavior, XP-C fibroblasts do not have the canonical traits of CAFs isolated from sporadic CCs. Whether the blockage of activation of cMET is sufficient to inhibit the promoting roles of XP-C fibroblasts on cancer cell propagation in vivo requires further studies.

Beyond HGF/SF, CAFs also directly stimulate SCC cell invasion by expressing various growth factors and cytokines, notably *TGFb1, GDF15, CXCL1, CXCL12, FGF7, IL6, OSM, LIF1, EGF,* and *IL1A* (Appendix A). However, in XP-C vs. WT fibroblast CMs, only *LIF1* and *SDF1a* mRNAs were slightly increased. Testing this in in vivo murine models treated with inhibitory cocktails targeting HGF, LIF1, SDF1a, and HGF/SF to see if they can limit SCC cell invasion in the SCC/XP context should answer this question. 

### 4.1. In Vivo Relevance

cMET may also be activated by physical stimuli such as membrane cell perturbations following either mechanical or undulatory stresses [9,25]. cMET has been shown to be activated in MDA-MB-231 breast carcinoma cells exposed to ionizing radiation (IR) or UVA radiation that induces both single-strand DNA breaks and the oxidation of cellular components. Also, as previously reported, we showed here that XP-C fibroblasts accumulate higher amounts of ROS compared to WT fibroblasts [26], notably after exposure to pro-oxidants (Figure 7), presumably leading to the modulation of gene expression, lipid modification, and mutagenesis. In this context, it is worth hypothesizing that further increases in HGF/SF levels in XP-C fibroblasts and downstream pathways may occur in vivo and lead to more aggressive SCC cells.

### 4.2. Molecular Mechanisms

How *XPC* mutations can change the repertoire of gene expression remains poorly understood and most researchers propose that XPC exhibits positive transcriptional functions in in vitro and ex vivo experimental settings, notably as a stimulating partner of basal transcription factor TFIIH [27,28,29]. Here, the absence of XPC was accompanied by an accumulation and not a decrease in HGF mRNA levels, protein production, and secretion. The vast majority of *XPC* mutations are deleterious [12,29,30]. However, XPC re-expression in *XPC*-transduced (NER proficient) XP-C fibroblasts failed to normalize the levels of HGF/SF mRNA along 2–5 cell propagations. The strong attenuation of XPC mRNA expression by Sh-RNA approaches in WT fibroblasts failed to increase the accumulation of *HGF/SF* mRNA (Appendix A). Thus, we can speculate that XP-C cells harbor epigenetic de-repression of some XPC “target genes”, notably *HGF/SF*, perhaps due to constitutive ROS accumulation (Appendix A). Consistent with this, Ho and colleagues reported the contribution of XPC in the control of the genetic methylation status in somatic pluripotent stem cells [31]. Longer overexpression or attenuation of *XPC* in XP-C primary cells ex vivo should confirm or reject hypotheses on the role of XPC in the control of gene transcription as well as epigenetic modifications. Together with our previous study showing that primary XP-C fibroblasts fail to express the CLEC2A activator of natural killer immune cells [10], the present findings relaunch discussions about the role of XPC in the regulatory mechanisms of gene transcription. 

Xenografts of cancer cells together with XP-C fibroblasts in the absence or the presence of HGF inhibitors may be expected to be a novel therapeutical approach for XP-C patients suffering from aggressive skin cancers (SCCs). Like sporadic CAFs in both spheroid and in vivo assays, XP-C fibroblasts aggregated in the proximity of cancers cells, suggesting close interactions between XP-C and cancer cells (Figure 6).

## 5. Conclusions

In conclusion, we identified a key role for XP-C fibroblasts in the invasion and tumorigenicity of SCC cells. The SCC cell invasion appears to be connected with the production of HGF/SF by XP-C fibroblasts and the subsequent activation of some cMET downstream pathways in SCC cells. A specific treatment targeting the HGF/cMET pathway in XP-C patients could prevent the development of aggressive carcinomas, as is already being applied in individuals from the general population [32].

## Figures and Tables

**Figure 1 cancers-16-03277-f001:**
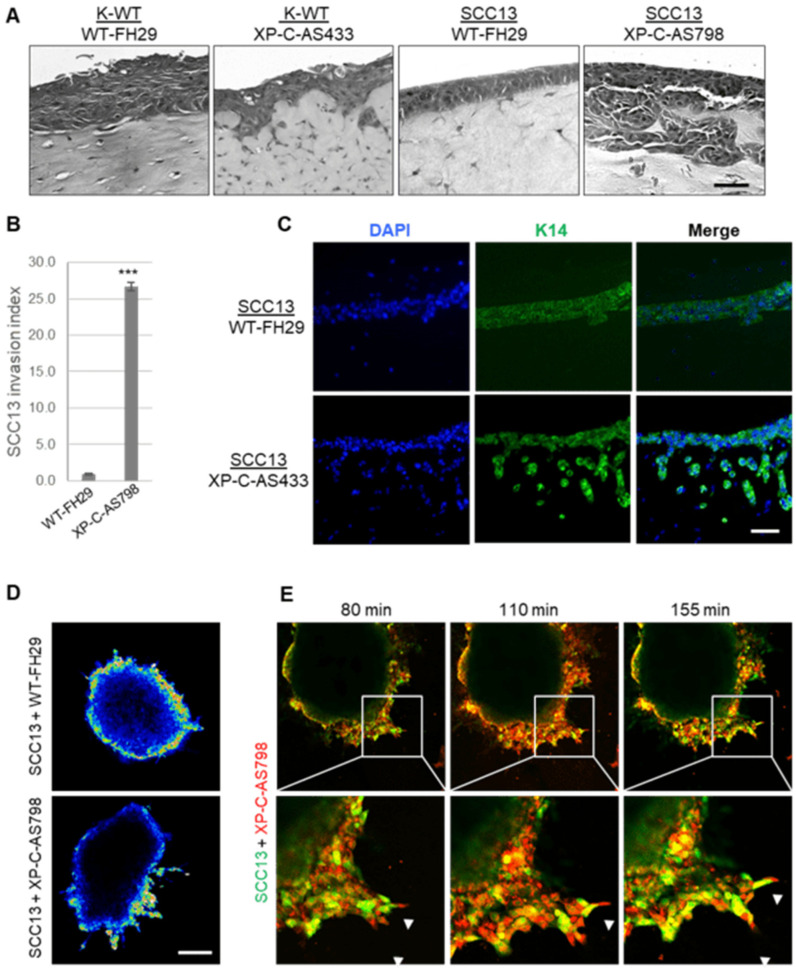
XP-C fibroblasts promote cancer cell invasion in organotypic skin cultures. (**A**) Representative images of H&E staining of sections from organotypic cultures generated from either WT keratinocytes (KTM1) or SCC13 cells together with either WT (FH29) or XP-C (AS433, AS798) fibroblasts embedded in dermal equivalents; bar: 50 µm. (**B**) Quantification of SSC13 cell invasion. Values are presented as the mean ± SD of three independent areas; *** *p* < 0.0001. (**C**) Immunofluorescence staining of K14 keratin (green) and nuclei (blue) show that invading cells are of epithelial origin; bar: 50 µm. (**D**) Spheroid assays composed of SCC13-GFP cells and either (red) WT (FH29) or XP-C (AS798) fibroblasts; bar: 250 µm. (**E**) Confocal microscopy images of spheroids showing SCC13-GFP cells and XP-C fibroblasts. Insets show higher magnification of spheroids. Arrowheads indicate the XP-C fibroblasts leading to invading protrusions. Bar: 40 µm.

**Figure 2 cancers-16-03277-f002:**
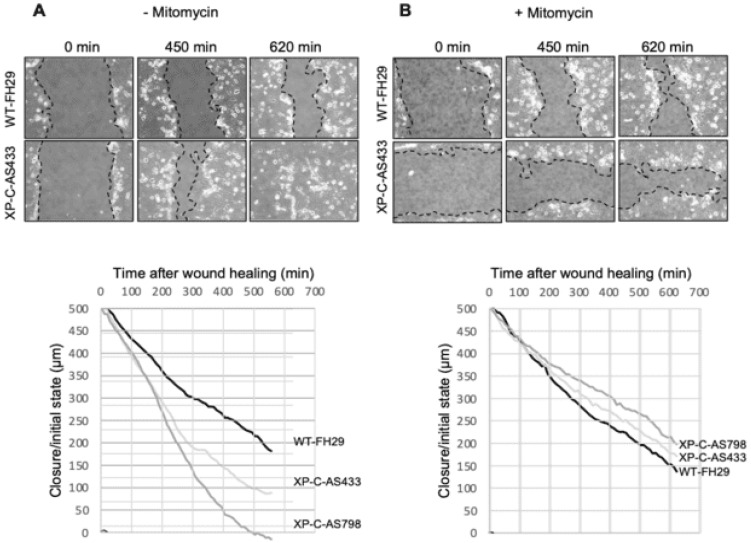
XP-C fibroblasts increase the rate of scratch closure of cancer cells ex vivo. Effects of either WT or XP-C fibroblast CM on SCC cells’ scratch closure without (**A**) or with MMC (**B**). Upper panels show representative micrografts from the time-lapse video microscopy of SCC13 cells’ scratch closure in the presence of either WT or XP-C CM at indicated times. Lower panels show the kinetics of SCC13 cells’ scratch closure in the presence of either 2 strains of XP-C CM (XP-C-AS433, XP-C-AS798) or 1 strain of WT fibroblast CM (WT-FH29). Dash lines indicate merging of the scratch. Note the substantially slower scratch closure kinetics of SCC13 cells in the presence of MMC (please also see Appendix A).

**Figure 3 cancers-16-03277-f003:**
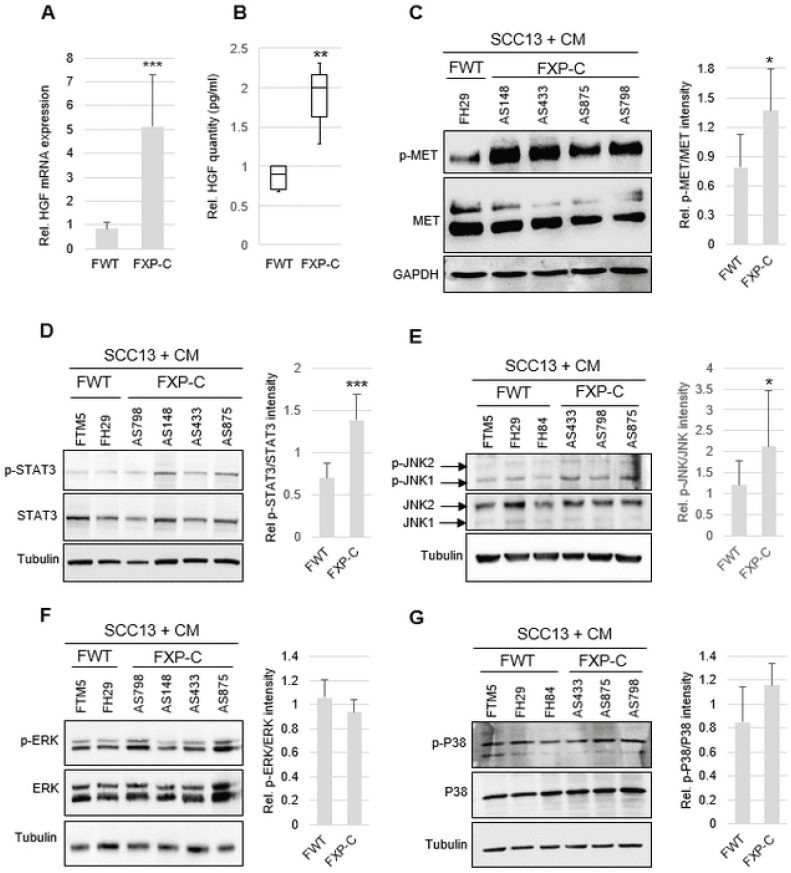
XP-C fibroblasts overexpress HGF/SF, leading to downstream activation of STAT3 and JNK c-MET pathways in SCC cells. (**A**) Accumulation of HGF mRNA was significantly higher in XP-C (*n* = 4) (FXP-C) than in WT (*n* = 2) fibroblasts (FWT). Values are presented as the mean ± SD of 4 independent experiments, *** *p* < 0.001. (**B**) HGF levels (ELISA) were significantly higher in XP-C fibroblast CM (*n* = 3) than in WT fibroblast CM (*n* = 3); ** *p* < 0.01. (**C**) Immunoblot analysis of cMET and p-MET in SCC cells treated with CM of either WT (FH29, FH84) or XP-C (AS148, AS433, AS875, AS798) fibroblasts. (**D**–**G**) Western blot analysis of phosphorylated forms of STAT3, JNK1,2, ERK1, 2, and P38 in SCC13 cells cultured in CM of WT (FTM5, FH29) or XP-C (AS148, AS433, AS875, AS798) primary fibroblasts. β-tubulin was used as the loading/transfer control. Right panel quantifies the ratio of phosphorylated to total protein amounts. Values are the mean ± SD of two independent experiments. *** *p* < 0.001, * *p* < 0.05.

**Figure 4 cancers-16-03277-f004:**
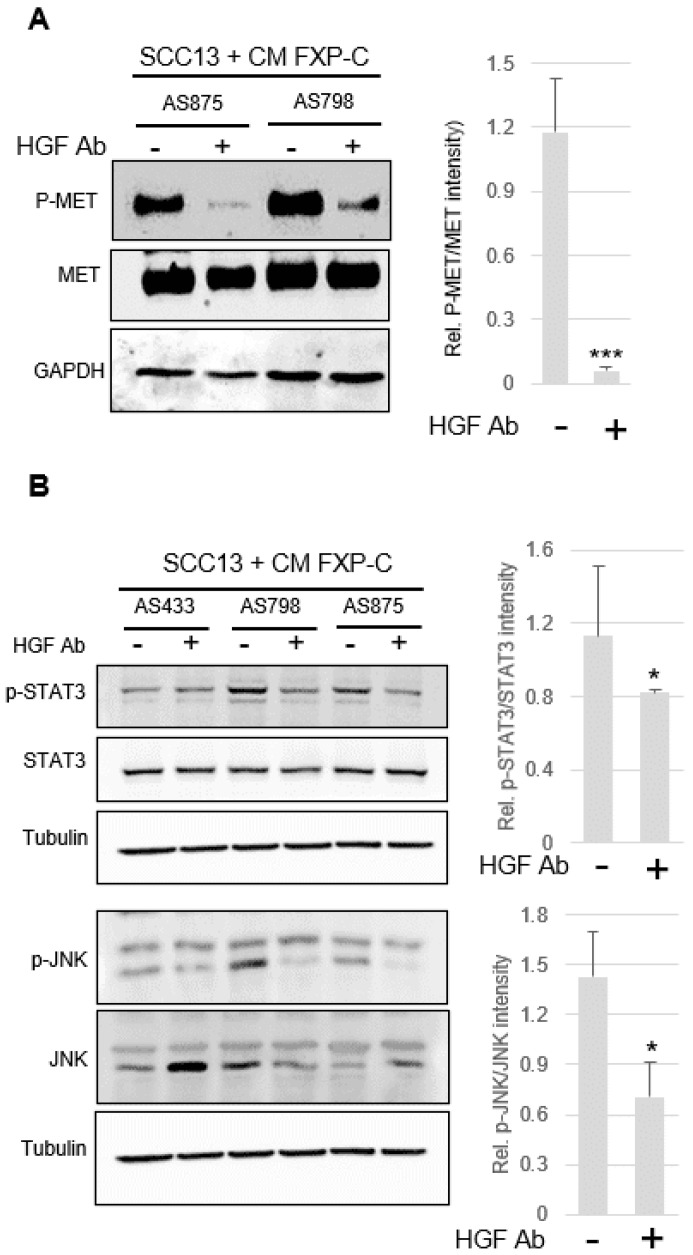
Blockade of cMET/HGF in culture media conditioned by primary XP-C fibroblasts reduces activation of JUNK and STAT3. (**A**) Western blot analysis of total cMET and p-MET levels in lysates of SCC cells cultured in XP-C fibroblast CM (AS875, AS798) ± HGF blocking antibody. GAPDH was used as the loading/transfer control. Right panel shows quantification of p-MET accumulation ± anti HGF-blocking antibody. (**B**) Western blot analysis of p-STAT3 and p-JNK levels in lysates of SCC cells cultured in the presence of XP-C fibroblast CM (AS798) ± HGF blocking antibody. Tubulin was used as the loading/transfer control. Right panel shows the quantification of the ratio of p-STAT3 and p-JNK to the total protein levels under the different conditions. Values are the mean ± SD. *** *p* < 0.001; ** p* < 0.05). For Figure 3 and Figure 4, raw images of blots are shown in Appendix A.

**Figure 5 cancers-16-03277-f005:**
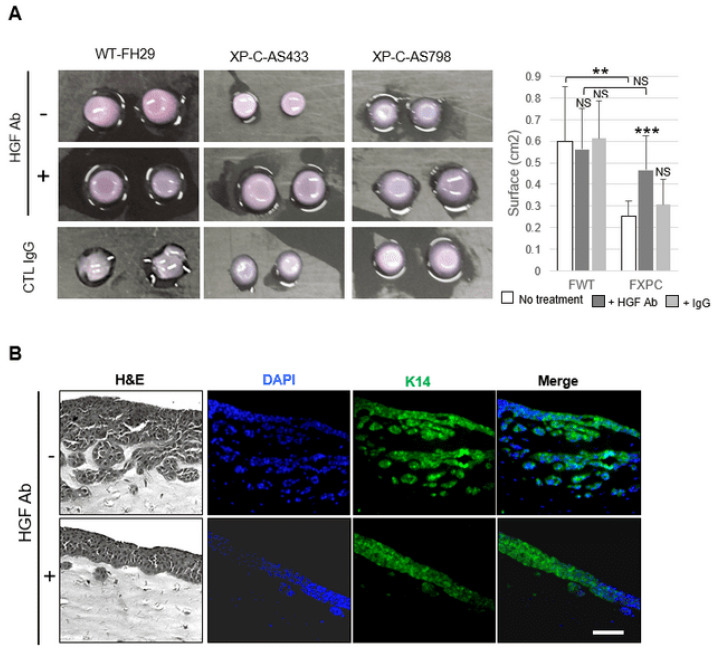
Contractile and invasive properties of cancer cells are normalized by HGF antibodies in XP-C CM. (**A**) Organoids containing either WT (FH29) or XP-C (AS433, AS798) fibroblasts ± HGF blocking antibody; quantitation is showed in the right panel. (**B**) Organotypic cultures composed of SCC cells and XP-C (AS798) fibroblasts ± HGF blocking antibody. Sections were stained using either H&E, DAPI (blue), or K14 antibody (green). Cancer cell invasion was significantly reduced by the HGF antibody; bar: 50 µm. Values are the mean ± SD. ** *p* < 0.01, *** *p* < 0.001.

**Figure 6 cancers-16-03277-f006:**
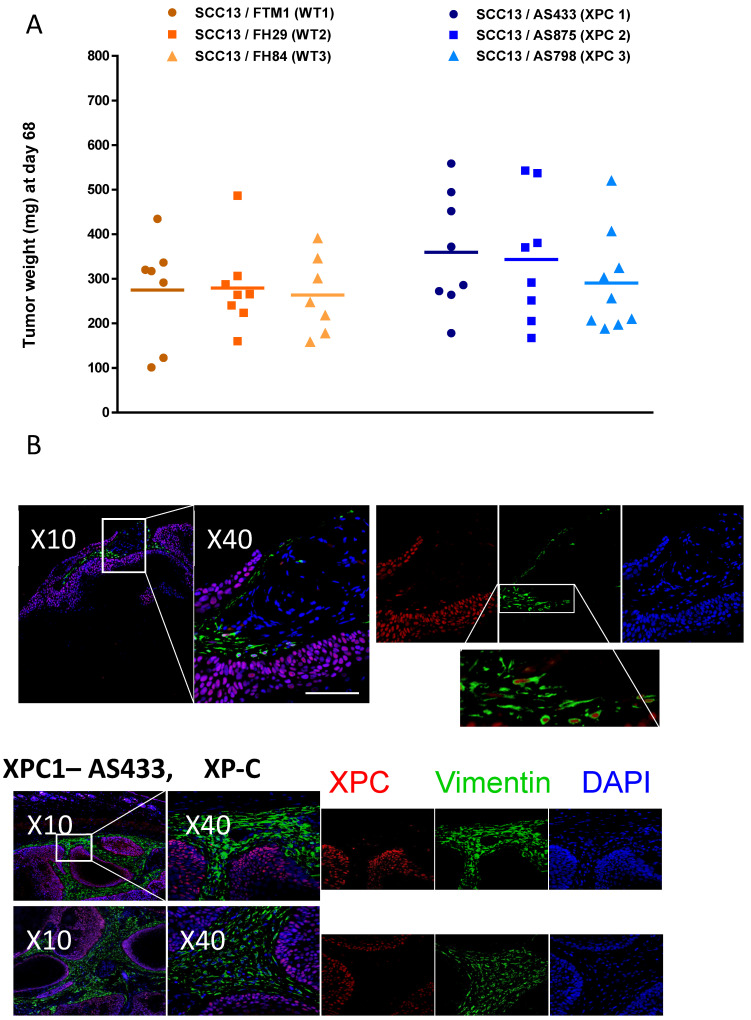
Contribution of XP-C fibroblasts to carcinoma cell invasion in vivo. (**A**) Numbers and tumor volume were measured in either absence or presence of XP-C fibroblasts. (**B**) In contrast to WT primary fibroblasts (FTM1, FTM5, FH84, FH29; representative image in upper panel), XP-C primary fibroblasts (AS 433, AS 875, AS 875, AS 673, AS 202; lower panel) accumulated in the vicinity of epithelial cancer cell invasions (representative image of graft sections from xenografts). Labeling colors are indicated; bar: 100 µm.

**Figure 7 cancers-16-03277-f007:**
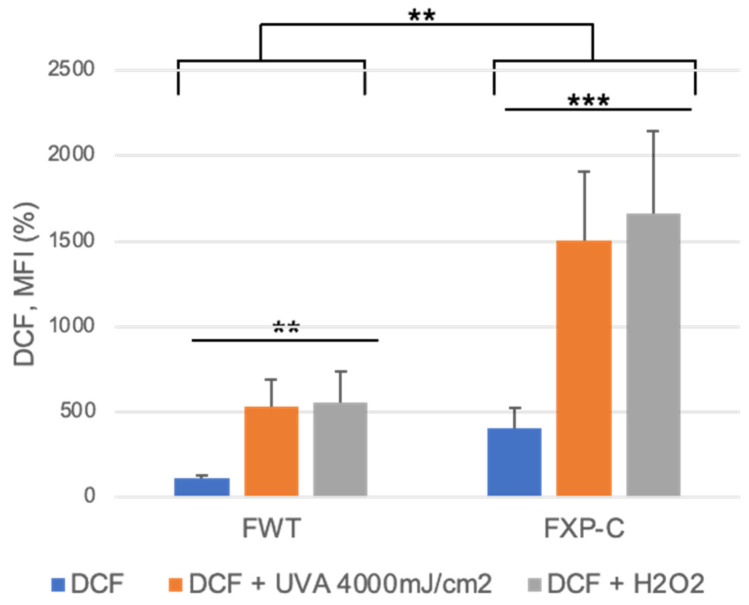
XP-C primary fibroblasts accumulate higher ROS levels than WT primary fibroblasts after treatment with either H_2_O_2_ or UVA, as indicated in the figure. Pool of 2 independent experiments using WT FH 84 primary fibroblasts as controls; WT, *n* = 2; XP-C, *n* = 3. Asterisks indicate statistical significance: ** *p* < 0.01, *** *p* < 0.001.

## Data Availability

The datasets generated and/or analyzed in this study are available upon reasonable request.

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
