# Peer review of "Xeroderma Pigmentosum Type C Primary Skin Fibroblasts Overexpress HGF and Promote Squamous Cell Carcinoma Invasion in the Absence of Genotoxic Stress"

_cancers, 2024, doi:10.3390/cancers16193277_

Round 1

Reviewer 1 Report (Previous Reviewer 2)

Comments and Suggestions for Authors

The authors have made a big improvement with this revised version of their manuscript. The inclusion of new data and the addition of more information in the main text and in the Material and Methods section has been beneficial for their work.

Author Response

Thank you for your comment.

Reviewer 2 Report (Previous Reviewer 3)

Comments and Suggestions for Authors

Having read the manuscript I have the following comments.  Firstly I would like to thank the authors for this revised version where many of the comments made previously have been addressed.  However there are some areas of the manuscript that need to be revised, namely

1. The authors claim they have inserted a table containing the abbreviations used in this manuscript, there is no table shown in the manuscript, please insert it.

2. Section 2.5, please supply details of the antibodies used in this study where were they from and dose for each one used?   

3. L116, NaCl is not written NACL please correct.

4. L124, what do you mean 50 µg de proteins loaded?, please correct.

5. L334-339, the comments made by the authors are not supported by research data, so this should either be supported with referenced studies or removed.  As mentioned previously the authors are guessing what is happening without having supporting information to substantiate this claim.

6. Fig S5 should be incorporated into the main body of the manuscript and not left in supplementary material.  Please add this to the main text of the manuscript.

Comments on the Quality of English Language

The manuscript is much improved from the previous version. However there are some sections where the English grammar is not clear.

Author Response

Having read the manuscript I have the following comments.  Firstly, I would like to thank the authors for this revised version where many of the comments made previously have been addressed.  However there are some areas of the manuscript that need to be revised, namely

1. The authors claim they have inserted a table containing the abbreviations used in this manuscript, there is no table shown in the manuscript, please insert it. 

- Done

2. Section 2.5, please supply details of the antibodies used in this study where were they from and dose for each one used?

-Done

3. L116, NaCl is not written NACL please correct.

Done

4. L124, what do you mean 50 µg de proteins loaded?, please correct.

I might be wrong, but I did not understood the request; I checked the revised MS text but did not find “mean 50 µg de proteins loaded.”

5. L334-339, the comments made by the authors are not supported by research data, so this should either be supported with referenced studies or removed.  As mentioned previously the authors are guessing what is happening without having supporting information to substantiate this claim.

I do not agree with the reviewer on this matter. I think that the sentence "Whether in vivo murine models treated with inhibitory cocktails targeting HGF, LIF1, SDF1a and HGF/SF can limit SCC cell invasion in the SCC/XP context should answer this multiple-choice question " is to keep in the text as it is, except if inappropriate English use has been detected.

6. Fig S5 should be incorporated into the main body of the manuscript and not left in supplementary material.  Please add this to the main text of the manuscript.

Done: figure S5 is now figure 7 included in the text core (discussion section).

Comments on the Quality of English Language

The manuscript is much improved from the previous version. However, there are some sections where the English grammar is not clear.

I have no problem to see my MS edited for the English language if this can improve it further.

This manuscript is a resubmission of an earlier submission. The following is a list of the peer review reports and author responses from that submission.

Round 1

Reviewer 1 Report

Comments and Suggestions for Authors

In the present manuscript, it indicated that the Xeroderma pigmentosum (XP)-C fibroblasts promote the Squamous Cell Carcinoma (SCC) Invasion ex vivo and in vivo through over expression of the Hepatocyte Growth Factor (HGF). However, the manuscript is not well-written. I recommend that this paper accepted after minor revision.

1. In introduction, it is not sufficient to understand the background of this research. It is better to add the more explanation about the relationship between the XP-C fibroblasts and SCCs.

2. The authors mentioned XP-C fibroblasts promote the invasion with no genotoxic stress. However, there is no result about the genotoxic stress.

3. The ratio of JNK1/JNK2 protein was different between Figure 3E and Figure 4B.

4. Although the size of the organoids containing XP-C fibroblasts was smaller than that containing WT in Figure 5A, the tumor weight was almost same in Figure 6A. It should explain about the results.

5. For abbreviated characters such as CC, SCC, ECM, FAD and CAF, it should write the characters before the abbreviation.

6. In Line 64, there is an abnormal line break.

7. In Line 104, change 0,5% to 0.5%.

Reviewer 2 Report

Comments and Suggestions for Authors

Al-qaraghuli et al., focused on Xeroderma Pigmentosum (XP) disease, a severe rare recessive disorder characterized by ultraviolet-induced DNA lesions repair deficiency. Many XP patients are affected by skin carcinoma and melanoma in the childhood ( <8 years old). The authors in previous studies found that primary XP fibroblasts isolated from healthy-non-photo exposed-patient skin, negatively control the extracellular matrix and the activation of the innate immune system. In the current manuscript, the authors described that XP group C primary fibroblasts in the absence of genotoxic attacks play a key role in cancer cell invasion ex vivo and in vivo through the over expression of the Hepatocyte Growth Factor/Scatter Factor (HGF/SF). Using inhibitors of the HGF/SF signaling pathway the authors were able to counteract the effects of XP fibroblasts on the growth of cancer cells, suggesting new therapy avenues in the care of XP patients.

The article may potentially have clinical relevance. However, it is not always clear and lacks information.

Material and Methods are not adequately described. It would be more informative if the authors could provide a brief description for each subsection and avoid the reader to be referred to the Supplemental Section. Abbreviations should be opened when introduced for the first time. I would suggest to move Table S1 in the main manuscript. How did the authors select the cells to be used in the study? Why did not the authors focus on fibroblasts from young patients? What do the authors mean by “organotypic” cultures? Please, provide with more information.

The authors in Figure 2 described scratch closure of cancer cells in vivo. I have some concerns about the cells and the method itself. 1) Are WT-FH29 and XP-C-ASA33 both cancer cells? 2) To ensure true detection of migration, pre-treating with mitomycin C (mit C), a DNA synthesis inhibitor, is essential. I don’t understand why the authors included “results” from the same experiment without mit C. Please, remove Fig.2A or provide with a justification of this panel.

The authors claim that “for the first time they report that primary fibroblasts isolated from non-photo-exposed healthy skin from XP-C patients overexpress HGF/SF at significantly higher levels than WT fibroblasts ex vivo”. It is bizarre that the data is in the Supplemental Section and to me it does not look that convincing.

At page 11, lines 280-285: The authors should support this claim with data.

The authors should support their claims by using specific inhibitors for HGF/SF. Perhaps, I missed this piece of information.

Comments on the Quality of English Language

Minor English editing is required

Reviewer 3 Report

Comments and Suggestions for Authors

Having read the manuscript I have the following comments:

1.  The manuscript has many poorly explained abbreviations, please list a table with all the abbreviations used within.

2.  Please revise the entire materials and methods section, as it is poorly written and lacks data such as replicate samples etc.

3. Please note that a full stop does not represent a multiplication symbol, ie. L72 15.103 should be written as 15x103.  Please correct all errors, do not list 90 min as 1h30min, express it in min only.  Use the symbol ± instead of  +/- .

4.   The standard of English used in the manuscript needs revision, please have a native English speaker revise the manuscript.  Latin words such as in vivo etc should be written in italics.  Please write in past tense, and the cells are spun not spined (L67).

5. L135 what do you mean the text continues here?

6. L144 It is  "Time-lapse" not "Laps-time", please correct.

7. Fig 1B does this data represent 3 areas of the one slide or data averaged from 3 separate slides, please clarify.

8. Fig 2 is poorly explained and should not have - signs on the Y axis.  You refer to gap closure and using a negative sign means widening of the gap not closing.  Please correct.

9. In Fig 3 what does FWT or FXP-C mean?  No explanation is given as to what these abbreviations represent.

10. Fig 3A data from what cells are used in this histogram?  Were these CM-treated cells or not?

10. The stats shown in Fig 3C & E are questionable as overlapping error bars for SD measurements usually do not suggest a significant difference, can you please clarify this.  

11.  Where is the information on Fig 3D-G shown in the legend?  It is not present after L200.

12. How many replicates are there for the data shown in Fig 4-6?

13. The image shown in Fig 4B do not appear to be from the same blot as the p-JNK bands do not align with JNK or Tubulin.  

14. Are the images shown in Fig 5A correct, adding HGF Ab should reduce the size of the organoids not increase them.   can you please elaborate on why adding neutralising Ab to HGF should increase the size of the tumour spheroid.

15.  insert larger images for Fig 6B these are too small to see what is in them.

16.  L280-285 this section is speculation as no direct measurements were made in this manuscript.  

17. Please include the data on ROS data (L291-3) in the main body of the manuscript.  

18. In the supplementary data why were organoids cultured in a 10% CO2 atmosphere.  

19. Why were the western bloods run using 8% SDS-PAGE gels, what voltage were the gels run at?  No information is given.  Please add this information.

20.  Supplementary Page 5 what is meant by the statement Gel 1:  etc.  It was not possible to follow what was written here.

21.  Supplementary Figures 5-6 are incorrectly labelled or lack what they say they contain.  Please ensure the correct data is shown in these figures.

22.  What does cFAD stand for, I could not find what this abbreviation represented.

Comments on the Quality of English Language

The standard of English used in the manuscript needs to be extensively edited and improved.  The grammar and use of abbreviations make it hard to follow in sections.  Revising the manuscript will considerably improve its readability.